# Robust whole slide image analysis for cervical cancer screening using deep learning

Shenghua Cheng[1,2,7], Sibo Liu[1,2,7], Jingya Yu[1,2,7], Gong Rao[1,2], Yuwei Xiao[1,2], Wei Han[1,2], Wenjie Zhu[3], Xiaohua Lv[1,2], Ning Li[1,2], Jing Cai[4], Zehua Wang [4], Xi Feng[5], Fei Yang[5], Xiebo Geng[1,2], Jiabo Ma[1,2], Xu Li[1,2], Ziquan Wei [1,2], Xueying Zhang[1,2], Tingwei Quan[1,2], Shaoqun Zeng[1,2], Li Chen[6✉], Junbo Hu[3✉] & Xiuli Liu [1,2✉]

Computer-assisted diagnosis is key for scaling up cervical cancer screening. However, current recognition algorithms perform poorly on whole slide image (WSI) analysis, fail to generalize for diverse staining and imaging, and show sub-optimal clinical-level verification. Here, we develop a progressive lesion cell recognition method combining low- and high-resolution WSIs to recommend lesion cells and a recurrent neural network-based WSI classification model to evaluate the lesion degree of WSIs. We train and validate our WSI analysis system on 3,545 patient-wise WSIs with 79,911 annotations from multiple hospitals and several imaging instruments. On multi-center independent test sets of 1,170 patient-wise WSIs, we achieve 93.5% *Specificity* and 95.1% *Sensitivity* for classifying slides, comparing favourably to the average performance of three independent cytopathologists, and obtain 88.5% true positive rate for highlighting the top 10 lesion cells on 447 positive slides. After deployment, our system recognizes a one giga-pixel WSI in about 1.5 min.

[1] Collaborative Innovation Center for Biomedical Engineering, Wuhan National Laboratory for Optoelectronics-Huazhong University of Science and Technology, Wuhan, Hubei, China. [2] Britton Chance Center and MOE Key Laboratory for Biomedical Photonics, School of Engineering Sciences, Huazhong University of Science and Technology, Wuhan, Hubei, China. [3] Department of Pathology, Maternal and Child Hospital of Hubei Province, Tongji Medical College, Huazhong University of Science and Technology, Wuhan, Hubei, China. [4] Department of Obstetrics and Gynecology, Union Hospital, Tongji Medical College, Huazhong University of Science and Technology, Wuhan, China. [5] Department of Pathology, Hubei Cancer Hospital, Tongji Medical College, Huazhong University of Science and Technology, Wuhan, Hubei, China. [6] Department of Clinical Laboratory, Tongji Hospital, Tongji Medical College, Huazhong University of Science and Technology, Wuhan, Hubei, China. [7] These authors contributed equally: Shenghua Cheng, Sibo Liu, Jingya Yu. ✉email: chenliisme@126.com; cqjbhu@163.com; xlliu@mail.hust.edu.cn

Cervical cancer is one of the most common cancers in women. In 2020, there were about 604,127 women diagnosed with cervical cancer worldwide, and 341,831 died of the disease[1]. Many studies show that periodic inspection can reduce the incidence and mortality of cervical cancer[2–5]. Traditional smear tests require doctors to read the slides under microscopes, and usually each slide has tens of thousands of cells. During the diagnosis process, the cytopathology staff need to spend a lot of time traversing all the cells and diagnosing the suspicious cells among them[6]. Therefore, the manual screening is very labor-intensive and experience-dependent, possibly resulting in low sensitivity and false negatives[7,8]. With the progresses of digital whole-slide image-scanning instruments[9] and computer image processing technologies[10], a lot of automated lesion cell recognition methods[11] are developed and bring hope for accurate and efficient computer-aided cervical cancer screening.

Traditional methods, mainly based on morphological and textural characteristics[12], generally consist of image segmentation, feature extraction, and cell classification. The image segmentation is used to segment the nucleus or cytoplasm through image histogram threshold, optical density measuring, and image gradient[13–16]. The feature extraction primarily focuses on the shape features and textural information of nuclei. Cervical cells are then classified by random forest, support vector machine, and artificial neural network[17,18]. The performance of such methods is highly dependent on the segmentation effect and feature engineering. Subject to the principles, traditional methods have low accuracy for distinguishing lesion cells with fuzzy classification boundaries and limited generalization for diverse cytology slides derived from staining and imaging. In order to solve this problem, some commercial systems such as BD FocalPoint Slide Profiler[19] and Hologic ThinPrep Imaging System[20] adopted a closed-loop strategy that integrates slide preparation, staining, imaging, and recognizing to ensure the accuracy and stability of the systems. In fact, they circumvented the generalization problem without solving it, which limits the wide use of the products especially in impoverished areas.

With the development of deep learning[21], convolutional networks (CNNs) have been applied to the identification of cervical lesion cells. Some studies have shown that CNNs improve the effect of nucleus segmentation[22,23], and others utilize image classification and object detection CNNs to directly identify lesion cells without traditional segmentation process[24–29]. Compared with traditional methods, CNN-based methods learn feature representations automatically and have better generalization potential. However, current CNN-based computer-assisted diagnosis algorithms are insufficient in WSI-level analysis, generalization for diverse staining and imaging, and clinical-level verification. Most of the methods mainly focus on the recognition of local lesion cells, lacking WSI-level diagnostic analysis. Even a few methods[30,31] analyzed the whole-slide cervical images on large-scale datasets, but they still did not solve the generalization problem in practical applications and clinical-level verification. The image volume and annotation number of existing public datasets are small, such as Herlev[32], ISBI14[33], ISBI15[34], and CERVIX93[35], and most of them are provided with image tiles instead of WSIs, which hinders the progress of WSI-level diagnosis analysis. In addition, inference speed of CNNs on giga-pixel WSIs is challenging. Consequently, it is still difficult to apply current CNN-based methods in clinical cervical cancer screening scenes.

To address the above challenges in terms of WSI analysis accuracy, generalization, and speed, here we propose a computer-aided diagnosis system for cervical cancer screening based on deep learning and massive WSIs. In the cytopathologists' diagnosis process, they usually scan the slides under a low-power microscope to find suspicious cells, and then further confirm them under a high-power microscope. Inspired by the strategy and considering the accuracy and speed, we design a progressive recognition method combining the low- and high-resolution WSIs. First, a CNN screens WSIs at low resolution (LR) to quickly locate the suspicious areas, and then these areas are further identified at high resolution (HR) by another CNN. Finally, the system recommends the 10 most suspicious lesion areas in each slide for further reviewing by cytopathologists. Besides recommending suspicious lesion areas, our system also evaluates the lesion degree of WSIs and gives a probability through developing a recurrent neural network (RNN)-based WSI classification model. The CNN image features of the top 10 areas are extracted and input to the RNN model to get the positive probability of WSIs. We integrate designed data augmentation in HSV color space, diverse data learning with group and category balancing, and hard sample mining to achieve high accuracy and good generalization of our system. We train and validate our system on patient-wise 3,545 WSIs and 79,911 annotations from five hospitals and five kinds of scanners. On multi-center independent test sets of 1,170 patient-wise WSIs, we achieve 93.5% specificity and 95.1% sensitivity for classifying slides. For the most confusing 121 WSIs of them, we achieve 50.0% specificity and 74.6% sensitivity, closely equivalent to the average level of three independent cytopathologists. The recommended top 10 lesion cells on 447 positive slides have an average true positive rate (TPR) of 88.5%. Compared with the current Hologic ThinPrep Imaging System, our system has a higher TPR of recommended cells and is more robust to staining and imaging style. When deploying the system with C++, multi-threading and TensorRt[36] are used to accelerate image processing and forward inference. Our system recognizes one giga-pixel WSI in about 1.5 min using one Nvidia 1080Ti GPU. This speed ensures a good user experience in clinical applications and provides the possibility of real-time augmented reality under microscopes.

In short, our work establishes a WSI-level analysis system for cytopathology screening according to cervical slide characteristics of sparsely-distributed and tiny-scale lesion cells and gives an effective demonstration of using deep learning to solve the bottleneck problems of current cervical screening methods. The extensive validation experiments demonstrate that our system can be used for effectively grading slides and recommending top-ranked lesion cells and reducing the workload of cytology screening staff. We believe our robust WSI analysis system would act as an effective cytology screening assist and help accelerate the popularization of cervical cancer screening.

## Results

**System architecture**. Our progressive recognition system consists of the LR model, HR model, and WSI classification model, as shown in Fig. 1. The LR model is designed to quickly locate suspicious lesion areas at LR. The HR model is to identify the lesion cells and recommend the top 10 lesion cells at HR. The WSI classification model uses an RNN to integrate the CNN image features of the top 10 lesion cells, and outputs the positive confidence of WSIs. The LR model and HR model are both based on ResNet50[37]. For the LR model, we modify the fully connected layer of original ResNet50 and add a semantic segmentation branch for generating a rough location mask (Supplementary Fig. 1). Thus, the LR model can screen WSIs and locate the suspicious lesion areas. The semantic segmentation branch is constructed with residual blocks of dilated convolutions. The LR model accepts an image tile of $512 \times 512$ pixels (0.486 μm/pixel) as input and outputs a lesion probability and a location heatmap (Supplementary Fig. 2). Afterwards, for the areas with a

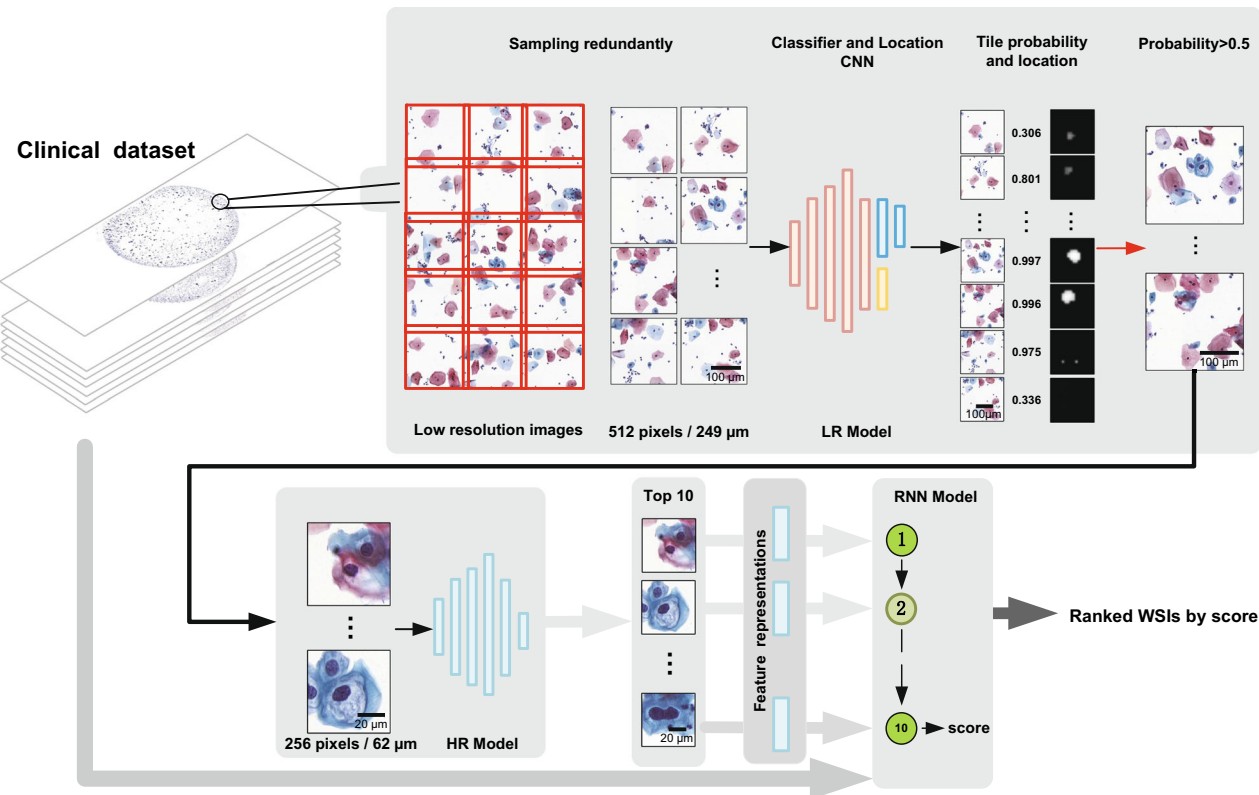

**Fig. 1 The proposed cervical cancer aided screening system.** Our system consists of WSI redundant division, LR model, HR model, and RNN model. The LR model takes a divided image tile of 512 × 512 pixels (0.486 μm/pixel) as input and outputs a lesion probability and a location heatmap to identify and locate the suspicious lesion areas on WSIs. The HR model takes an image tile of 256 × 256 (0.243 μm/pixel) cropped according to the location heatmap as input and outputs a new lesion probability. The RNN model integrates the HR model image features of the top 10 lesion cells and outputs positive probabilities of WSIs. The clinical dataset images in this figure were created by us.

probability higher than 0.5 predicted by the LR model, we perform some morphological operations on the corresponding location heatmap to generate the location mask. A cropped image tile of 256 × 256 (0.243 μm/pixel) according to the location mask is input to the HR model and a new lesion probability is obtained. Finally, all identified lesion cells in WSIs are sorted by lesion probabilities, and the top 10 typical lesion cells are recommended for cytopathologist reviewing. Further, the RNN model integrates the CNN image features of the recommended top 10 lesion cells to classify WSIs. For each lesion cell tile, 2,048-dimensional features are extracted by the HR model. Then the total 10 × 2,048-dimensional features are input to the RNN model, and positive probabilities of WSIs are output.

**Multi-center WSI datasets**. To assess the robustness and clinical applicability of our system, we collected 12 groups of datasets from five hospitals and five kinds of imaging instruments (see "Dataset sources" in "Methods"), which are referred to as groups A–L (Fig. 2a). These 12 datasets include 1,467 (41.4%) positive WSIs and 2,073 (58.6%) negative WSIs with 79,911 annotated lesion cells by a consensus of three cytopathologists. Each WSI represents one unique patient. The 12 datasets show completely different image styles of staining and imaging characteristics (Fig. 2b) and we quantified the difference in their numerical distributions (Fig. 2c). Groups A–D are used for training our system. Groups E–L are treated as a completely independent test set to evaluate the generalization of our system. Groups A–D are randomly divided into training set, validation set, and test set with a slide-wise ratio of 8:1:1 (Fig. 2d). WSIs of all groups are scanned under ×20 or ×40 magnification microscopes. We

uniformly interpolate them to 0.243 μm/pixel in data preprocessing, since the different resolutions of various imaging instruments.

In order to verify the effect of our recognition system in practical applications, we invited three cytopathologists to evaluate the prediction results of 1,170 slides in groups E and F. Groups E and F are independent from the training data with new styles and thus are suitable for clinical-level experiments. We performed the below assessments: slide level accuracy, tile level accuracy, and true positive rate of recommended top 10 lesion cells. All the skilled cytopathologists for annotation have been trained and certified by the Chinese Society for Colposcopy and Cervical Pathology.

**Assessment at the slide level**. To assess the effectiveness of our system at the slide level, we compared the RNN classifier and cytopathologists in classifying WSIs on the independent groups E and F of 1,170 slides. Figure 3a shows the ROC (receiver operating characteristic) curve of our system for classifying positive and negative slides, achieving 93.5% specificity and 95.1% sensitivity with 0.979 AUC (the area under ROC). The most confusing 121 slides of the slides were classified by the RNN and the three cytopathologists. Each red dot in Fig. 3b refers to 1-specificity and sensitivity of a cytopathologist's interpretation result. Our system achieves 50.0% specificity and 74.6% sensitivity with 0.647 AUC, which is comparable with the average level of cytopathologists. In addition, our system processes one giga-pixel WSI in about 1.5 min after deploying on a single GPU card, much faster than manual slide reading time.

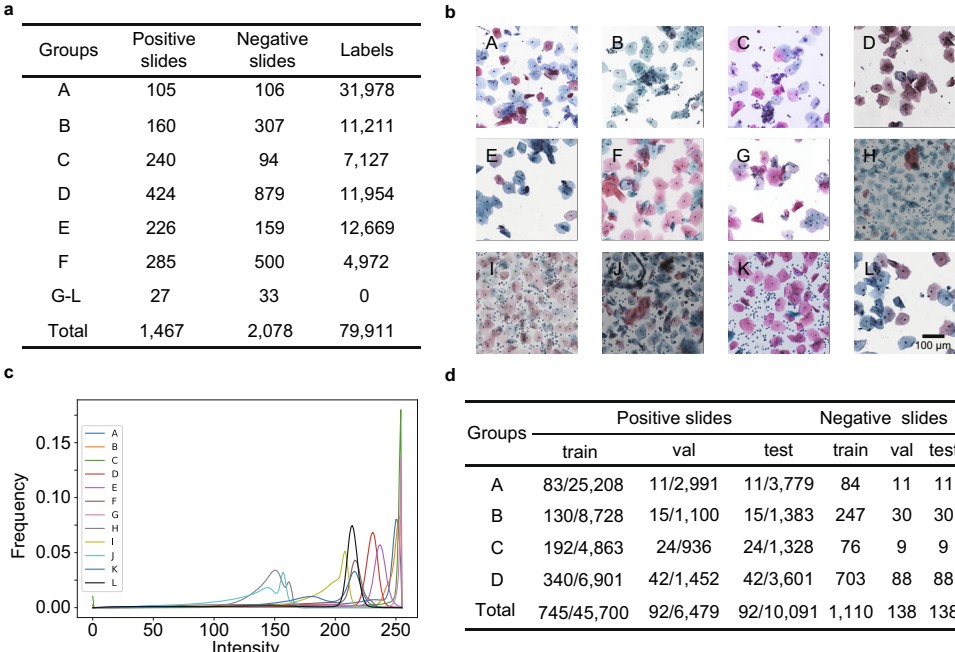

**Fig. 2 Overview of multi-center WSI datasets. a** The collected 12 groups of WSI datasets from five hospitals and five kinds of scanners. There are in total 3,545 WSIs with 79,911 annotated lesion cells by a consensus of three cytopathologists. **b** Cervical image instances of the 12 groups, showing diverse image styles of staining and imaging characteristics. **c** The numerical distributions of the 12 groups in the value channel of HSV space (hue, saturation, value), further confirming the difference of image styles. **d** The division of training set, validation set, and test set with a slide-wise ratio of 8:1:1 on groups A–D, which are used for training our system.

To test the performance of our system in the case of HPV testing first or in combination with cytology, we further analyzed the cervical cell slides from 395 HPV-positive patients (cytology positive 169, negative 226). We achieved 81.9% specificity and 79.3% sensitivity with 0.890 AUC (Supplementary Fig. 3). The main reason for the decrease is that many of these HPV-positive and cytology-negative samples are accompanied by bacterial infections, which may increase the classification difficulty. Our system was designed for the general population of women, and there was no special training for HPV-positive slides. We conducted the trial with the read-made networks without retraining. Although the preliminary results are interesting, the classification effect is not enough. If the HPV-positive samples are trained purposefully in the future, the classification effect is expected to be improved.

**Analysis of false-positive and false-negative slides**. From the frequency histogram of slide classification scores of groups E and F (Fig. 4), our system produces 0.8% false-negative slides (the slide score threshold value is 0.5), all of which were confirmed as ASC-US slides (atypical squamous cells of undetermined significance). As we know that cervical cytology ASC-US slides and part hard negative slides are confusable; thus, it is acceptable to misjudge a small number of ASC-US slides. Meanwhile, our system produces 26.3% false-positive slides, which is in line with the original intention of cervical cytology computer-aided diagnosis. These false positives will be further reviewed by cytopathologists. Further, our system achieves 49.3% specificity while retaining 100% sensitivity, indicating that 49.3% negative slides in the test groups E–F can be excluded. For different test groups, this ratio value may vary since that it depends on the lowest score of positive slides. The results indicate that our WSI analysis system can be applied for prescreening part completely normal slides and reducing the workload of cytotechnologists.

**Assessment at the tile level**. To evaluate the difference between our system and three cytopathologists at the tile level, we randomly selected 1,018 positive tiles and 3,047 negative tiles with a size of $256 \times 256$ (0.243 µm/pixel) from the test data of groups E and F. As shown in Fig. 3c, the ROC curve describes the performance of our system, and each red dot represents 1-specificity and sensitivity of a cytopathologist's classification result. Our system achieves 95.3% specificity and 92.8% sensitivity with 0.979 AUC better than the average level of cytopathologists.

**Assessment on the recommended top 10 lesion cells**. Three cytopathologists evaluated the recommended top 10 lesion cells on 447 positive slides in groups E and F. The average true positive rates of top 10 recommended cells evaluated separately by three cytopathologists are 96.8%, 88.0%, and 80.6% (Fig. 3d), with an average value of 88.5%, and the average true positive rate of top 20 recommended cells is 85.0%. For some atypical positive slides with a few lesion cells, such as ASC-US, the recommended true positive cell number may be very low as shown in the box plot (Fig. 3d). In this case, our system employs voting the evaluated results of three cytopathologists, and the final result shows we do not miss any positive slide, i.e., positive slides at least have one true lesion cell in the recommended top 10 or top 20 cells. Figure 4 shows the recommended cells of slides with different classification scores. For high-risk slides, our system recommended typical lesion cells such as koilocytotic cells or hyperchromatic cells with large nucleus and irregular nuclear membrane. For medium-risk slides, some suspicious cells with slightly large or deep-stained nucleus were recommended. No typical lesion cells were recommended on low-risk slides. Our system can accurately recommend a few top-ranked lesion cells, allowing cytopathologists to focus on these suspicious areas.

**Comparison of our system and Hologic ThinPrep Imaging System on recommending top lesion cells**. To further verify the

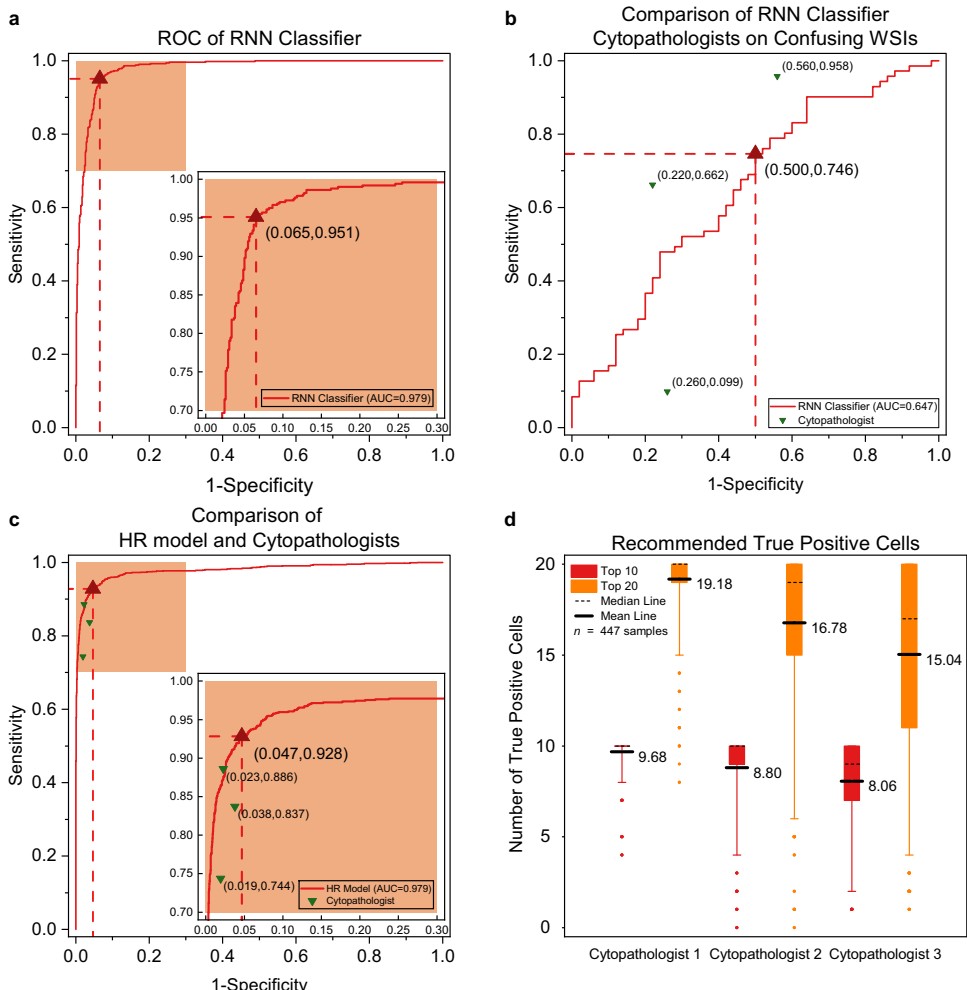

**Fig. 3 Clinical-level experimental results. a** The ROC of the RNN model for classifying the slides of groups E and F ($n = 1,170$). **b** Comparison of the RNN model and three cytopathologists at the slide level ($n = 121$). The most confusing 121 slides of the 1,170 slides were classified by the RNN model and the three cytopathologists. Each green triangle refers to 1-specificity and sensitivity of a cytopathologist's result. **c** Comparison of the HR model and the three cytopathologists at the tile level. Randomly selected 1018 positive test tiles and 3,047 negative test tiles with a size of $256 \times 256$ ($0.243\,\mu m$/pixel) from groups E and F ($n = 4,065$ tiles) were simultaneously classified by the HR model and three cytopathologists. **d** The average true positive numbers of the recommended top 10 and top 20 lesion cells on positive slides in groups E and F ($n = 447$), evaluated by three cytopathologists separately. The boxes indicate the upper and lower quartile values, the whiskers indicate the 95% and 5% quantile values, the middle bold solid line and dotted lines indicate the mean and median values, and the scatter dots indicates outliers.

effectiveness of our aided screening system, we compared it with the Hologic ThinPrep Imaging System (referred to as TIS). The test data are cervical cytology samples from 58 positive patients in Maternal and Child Hospital of Hubei Province equipped with TIS. First, 58 glass slides were prepared from the 58 samples, stained, imaged, and identified by TIS in the hospital. Twenty-two suspicious fields of view were recommended by TIS on each slide. Then, we used another instrument (Shenzhen Shengqiang Technology Ltd with $0.180\,\mu m$/pixel under $\times40$ magnification) to scan the 58 glass slides, and we used our system to recommend the top 20 suspicious cell regions (about $60 \times 60\,\mu m^2$, far smaller than TIS' fields of view) for each slide. We asked three cytopathologists to evaluate the results recommended by TIS and our system at the same time. The statistical results in Fig. 5 show that the true positive rate of our system is higher than that of TIS. Notably, TIS can only work under the closed-loop strategy of preparation, staining, imaging, and recognition, while our system is robust to staining and imaging of various sources.

**Importance of designed data enhancement, hard sample mining, and diverse data learning**. We conducted a set of ablation experiments to demonstrate the importance of designed data enhancement in HSV color space, hard sample mining, and diverse data learning with group and category balancing (see "Methods"). We used the three learning strategies to train a series of control high-resolution models step by step, and gave the classification accuracies on the test sets of groups A–F. Notably, the ratio of positive and negative tiles in the test set is 1:1. The ablation experimental model configs and results are provided in Fig. 6a. To evaluate model generalization, we treated groups E–F as the independent test data and showed the ROC curves of these control models on groups E–F in Fig. 6b. According to the results, with the designed data enhancement and hard sample mining, performance of the enhanced and mined models on groups E–F made great progress with the AUC value increase of 0.138 and 0.072. The results indicate that our designed data enhancement and hard sample mining strategies are effective for improving model generalization and accuracy. Further, as more groups of

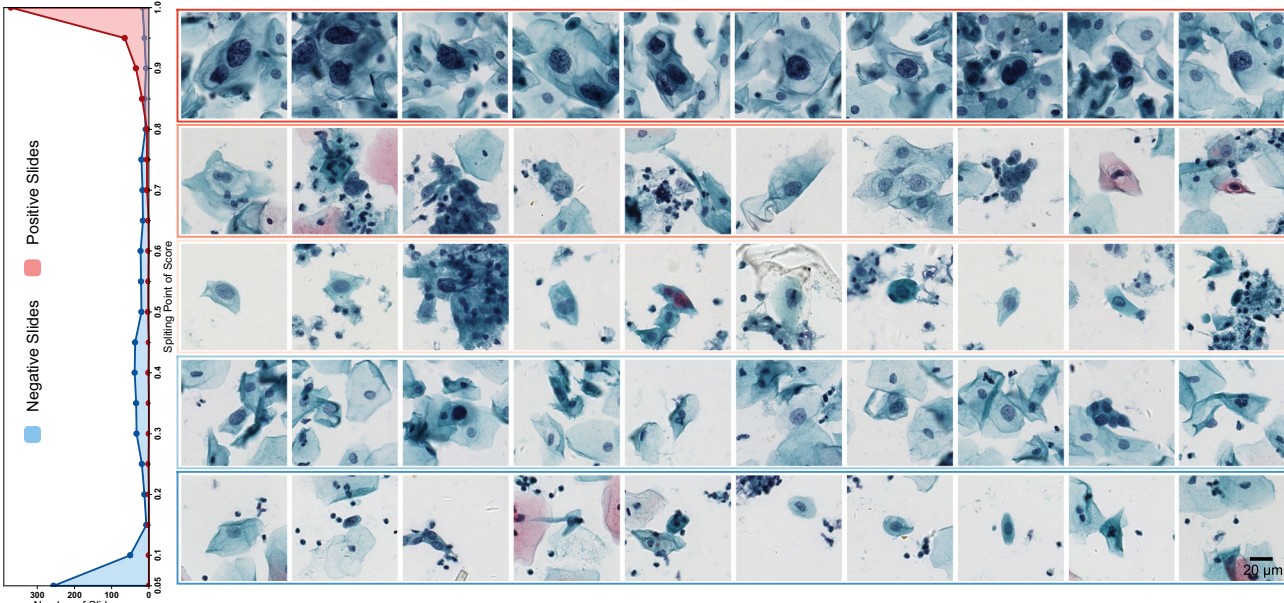

**Fig. 4 The recommended top 10 lesion cells of WSIs with different classification scores.** The left subgraph is the frequency histogram of slide scores from 0 to 1 with an interval of 0.05 in groups E and F (n = 1,170). The right subgraph is the recommended top 10 lesion cells of slides with different scores.

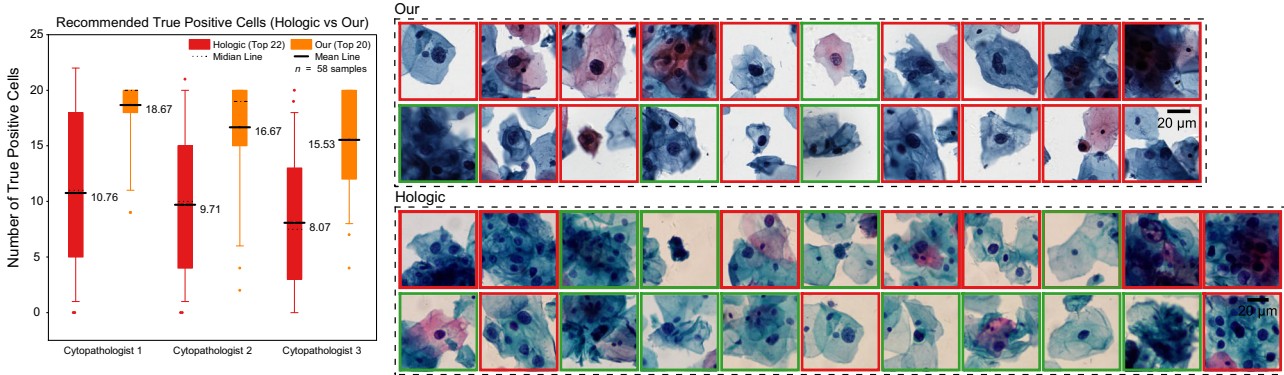

**Fig. 5 Comparison of our system and Hologic TIS on recommending top lesion cells.** The histogram on the left shows the true positive cell number of our system's top 20 cell regions (about $60 \times 60\ \mu m^2$) and TIS' top 22 fields of view (far greater than $60 \times 60\ \mu m^2$) on 58 positive slides (n = 58). The boxes indicate the upper and lower quartile values, the whiskers indicate the 95% and 5% quantile values, the middle bold solid line and dotted lines indicate the mean and median values, and the scatter dots indicate outliers. The subgraph on the right is recommended cells of one positive slide. Notably, the most suspicious cells were cropped from TIS' top 22 fields of view according to cytopathologists' evaluation results. The cells with the red border line were evaluated as true positive cells by cytopathologists.

training datasets were used, the AUC values of the mined, baseline, and HR models increased gradually from 0.808 to 0.983. The results indicate that the diverse data learning of multiple groups with different styles is important for model generalization.

**Generalized and rich feature representations of our models**. We analyzed the alignment of the features of high-resolution models between different groups of data by feature visualization. The dimension-reduced features of the original, enhanced, mined, and baseline models by t-SNE[38] on groups A, B, E, and F are shown in Fig. 7a. For the models, groups E and F are independent test data. From the original model to the baseline model, features of positive and negative tiles are gradually separated. Further, the features are gradually aligned between groups A-B and groups E-F. The results indicate that the designed data enhancement, hard sample mining, and diverse data learning strategies improve the discrimination and alignment of features on unseen groups E-F.

We further analyzed the feature representations of the HR model on groups A–L in Fig. 7b. The tiles with high and low lesion probabilities from the 12 groups are clustered and well separated. Tiles corresponding to the far-right points are the typical lesion cells, including koilocytotic cells and hyperchromatic cells with a large nucleus and an irregular nuclear membrane. These lesion cells with different staining and imaging characteristics are clustered together and share similar features. Normal cells from different groups are clustered on the left points. At the junction regions are the suspicious cells with about 0.5 lesion probabilities. The suspicious cells generally contain a slightly large nucleus or deep-stained nucleus, but the degree is not enough. In addition, artifacts from staining and imaging may cause the suspicious cells. The results indicate that the learned features represent cervical lesion cell morphology well and the features are aligned between datasets with different staining and imaging characteristics. This is the key reason why our system has good generalization for unseen datasets of new styles.

**a**

| Models | Train Strategies | | | Train Groups | Accuracy of Test Groups | | | | | |
| | Data Enhance-ment | Hard Samples | Diverse Data Learning | | A | B | C | D | E | F |
|---|---|---|---|---|---|---|---|---|---|---|
| Original | ✗ | ✗ | ✗ | A | 0.842 | 0.809 | 0.745 | 0.611 | 0.571 | 0.564 |
| Enhanced | ✓ | ✗ | ✗ | A | 0.894 | 0.903 | 0.738 | 0.710 | 0.686 | 0.606 |
| Mined | ✓ | ✓ | ✗ | A | 0.913 | 0.916 | 0.773 | 0.788 | 0.664 | 0.784 |
| Baseline | ✓ | ✓ | ✗ | AB | 0.958 | 0.930 | 0.865 | 0.703 | 0.778 | 0.838 |
| HR model | ✓ | ✓ | ✓ | ABCD | 0.988 | 0.941 | 0.958 | 0.916 | 0.965 | 0.905 |

**b**

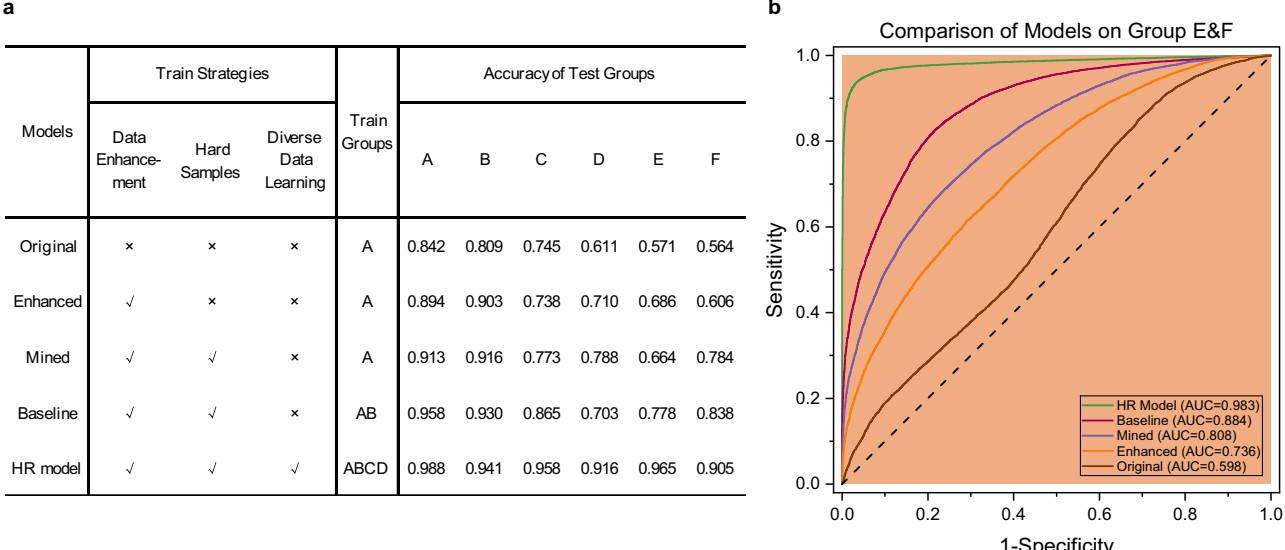

**Fig. 6 Importance of designed data enhancement, hard sample mining, and diverse data learning. a** The ablation experimental results about the designed three learning strategies of data enhancement, hard sample mining, and diverse data learning on the HR model. We used these learning strategies to train a series of control models step by step, and gave the classification accuracies on the test sets of groups A–F. **b** AUC-ROC comparison of the original, enhanced, mined, baseline, and HR models on groups E and F.

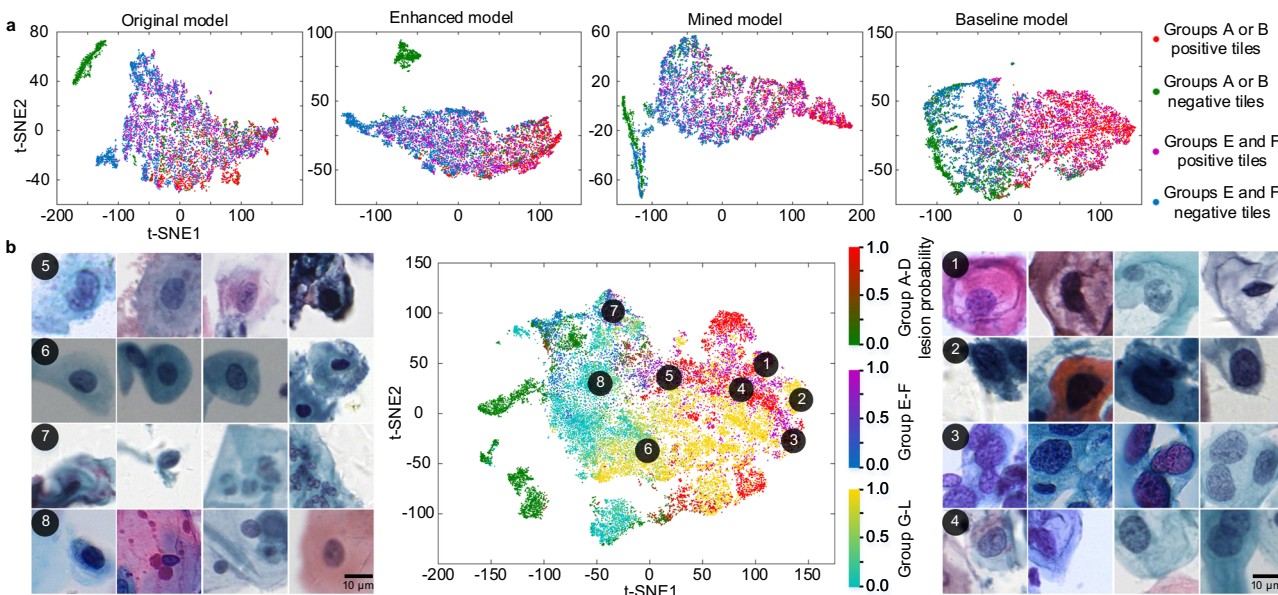

**Fig. 7 Generalized and rich feature representations of our models. a** Gradually aligned features between different groups of datasets. The last 2,048-dimensional features of the original, enhanced, and mined models were reduced to two dimensions using t-SNE on the test sets of groups A, E, and F and were plotted in the first three subgraphs respectively. The dimension-reduced features of the baseline model on the test sets of groups A, B, E, and F were plotted in the fourth subgraphs. In all, 1,000 positive test tiles and 1,000 negative test tiles in each group were randomly selected for the visualization. **b** The distribution of dimension-reduced features of the HR model on the 12 groups test sets and the interpretation of the feature representations by examining the corresponding cervical image tiles. In all, 2,000 test tiles in each group were randomly selected for the feature visualization. The different color bars refer to different groups and the gradient colors within a single color bar refer to the lesion probabilities of tiles.

## Discussion

In this paper, we propose a clinical-level-aided diagnosis system for cervical cancer screening based on deep learning and massive WSIs. Compared with the existing methods, our system has the following key advantages: (a) WSI-level analysis system rather than tile-level evaluation; (b) the integrated strategy of low- and high-resolution combination, data augmentation, diverse data learning, and hard sample mining for achieving high accuracy, good generalization, and fast speed of our system; (c) the

human–computer comparison verification at both tile and WSI levels to prove the effectiveness of our system; (d) the practical deployment with C++, processing a giga-pixel WSI in as short as about 1.5 min with one GPU.

The mean reported positive rates of cervical cytology screening are less than 10%[39,40], and that of the physical examination population is even lower. Therefore, if some negative samples can be distinguished, it should be a great aid to cytotechnology. In our study, the distribution of slide positive scores (0.8% false-negative slides) indicates

that our system has potential to exclude a considerable number of negative slides and reduce a lot of cytotechnologists' workloads. The 10/20 most suspicious areas on each slide with relatively high 88.5%/85.0% TPRs are given, which would help free doctors from the task of traversing and searching for suspicious targets and concentrate on the task of identifying recommended suspicious cells. For underdeveloped areas lacking cytopathology stuff, our system has important clinical and social significance for accelerating the popularization of cervical cancer screening.

In recent years, WSI analysis has been widely studied in various histopathology subspecialties[41–45]. These algorithms generally follow the below principle: first extract classification features or confidences of local tiles, then aggregate the local information to construct WSI-level feature descriptors, and finally classify the slides. The lesions of histopathology slides are region-level and have overall background information. Cytopathology slides show sparsely-distributed and tiny-scale lesion cells, and the cells are independent and short of overall information. These characteristics of cytopathology have brought challenges to accurate WSI analysis when directly transferring the histopathology methods. In this work, we propose a WSI-level analysis system for cytopathology screening according to cervical slide characteristics, and demonstrate its effectiveness in classifying cytology slides by the extensive validation experiments.

The diversities of slide staining and imaging in different hospitals greatly limit the utility of current automated cervical cell recognition algorithms; thus, model generalization is a key factor in the practicality. Our multi-center independent test datasets include differences in slide preparation (liquid-based preparation methods: membrane-based and sedimentation), dyeing schemes (fixing, clearing, and dehydrating), imaging magnification (×20 and ×40), imaging resolution (0.180–0.293 μm/pixel), and imaging color characteristics (Fig. 2b, c). The good results on these diverse data prove the generalization of our system and lay a foundation for the practicability in diverse data scenarios.

We consider that the basic task of the aided screening system is to reduce the workload of cytology staff by excluding low-risk slides and recommending a limited number of suspicious cells on high-risk slides for cytopathologist reviewing. The final precise diagnosis is up to cytopathology doctors, and this human–machine combination mode can reduce possible errors of artificial intelligence and ensure the accuracy of diagnosis[19,20]. Therefore, unlike the works of Lin et al.[30] and Zhu et al.[46], our system focuses on distinguishing positive and negative classes instead of fine subclasses at both cell and slide levels, such as ASC-US, ASC-H (atypical squamous cells cannot exclude high-grade squamous intraepithelial lesion (HSIL)), LSIL (low-grade squamous intraepithelial lesion), HSIL, and SCC (squamous cell carcinoma)[6]. Moreover, the definition of the subclasses is based on cell morphology and the boundaries are often fuzzy especially at the cell level, which will produce a lot of noises in the actual manual annotations and inconsistency between different cytopathologists.

In the future, we will focus on research about AI-enhanced portable microscopy and augmented reality microscopy to further expand our system. At present, professional but expensive scanners are still required, preventing the spread of cervical cancer screening in remote and underdeveloped areas. Thus, developing portable microscope-based cervical cancer computer-aided diagnosis is necessary. In addition, developing real-time augmented reality microscopes can provide friendly human–computer interaction for AI-assisted slide screening without changing the conventional working mode of cytopathologists.

## Methods

**Dataset sources**. All 12 groups of glass slides are provided by Maternal and Child Hospital of Hubei Province (referred as H1), Tongji Hospital of Huazhong

| Table 1 Dataset sources of the 12 groups of datasets. | | |
|---|---|---|
| **Groups** | **Hospital** | **Scanner** |
| A | H1 | S1—version 1 |
| B | H1 | S1—version 2 |
| C | H1 | S3 |
| D | H2—version 1 | S2 |
| E | H1 | S2 |
| F | H2—version 2 | S2 |
| G | H2—version 1 | S1—version 2 |
| H | H4 | S4 |
| I | H2—version 3 | S4 |
| J | H3 | S4 |
| K | H5 | S5 |
| L | H1 | S2 |

University of Science and Technology (referred as H2), Wuhan Union Hospital of Huazhong University of Science and Technology (referred as H3), Hubei Cancer Hospital (referred as H4), and KingMed Diagnostics Ltd (referred as H5). The slide acquisition is performed in accordance with the guidelines of the Medical Ethics Committee of Tongji Medical College at Huazhong University of Science and Technology. These glass slides are scanned into WSIs by the instruments from 3DHisTech Ltd with 0.243 μm/pixel under ×20 magnification (referred as S1), Shenzhen Shengqiang Technology Ltd with 0.180 μm/pixel under ×40 magnification (referred as S2), Wuhan National Laboratory for Optoelectronics-Huazhong University of Science and Technology with 0.293 μm/pixel under ×20 magnification (referred as S3), Huaiguang Intelligent Technology Ltd with 0.238 μm/pixel under ×20 magnification (referred as S4), and Konfoong Biotech Information Ltd with 0.238 μm/pixel under ×40 magnification (referred as S5). The details are shown in Table 1. Notably, version 1 and version 2 of the scanner S1 are different generations of instruments, and they have different imaging color characteristics. Version 1, version 2, and version 3 of the hospital H2 are different in slide preparation and staining scheme.

**Data annotating and screening**. Based on the TBS criteria[6], the cervical slides are annotated by six cytopathologists using Qupath[47] (v0.2.0) and a home-made semi-automatic online annotation software. Considering that it is a standard operating procedure in cytopathology when the diagnoses of the two pathologists are inconsistent, ask a senior doctor to make an interpretation[48,49]. In this work, the final annotations were produced by a consensus of three cytopathologists. We abandoned questionable annotations. Based on the TBS criteria[6], we classify all kinds of squamous epithelial cell abnormalities (including atypical squamous cells (ASC), squamous intraepithelial lesion (SIL), and SCC), and glandular epithelial cell abnormalities (including atypical endocervical/glandular cell, endocervical adenocarcinoma in situ (AIS), and adenocarcinoma) as positive labels, and classify various normal cellular elements, nonneoplastic findings such as nonneoplastic cellular variations, reactive cellular changes, and glandular cells' status post hysterectomy as negative label NILM (negative for intraepithelial lesion or malignancy). During the iterative learning of our system, we collected massive false negatives of various morphologies as negative annotations.

**Designed data enhancement**. Data enhancement is a common technique to expand the diversity of data distributions and improve the generalization ability of models. Based on the characteristics of cervical cell images, we designed a series of specific data enhancement in HSV and RGB color spaces to imitate the color distribution of various image styles derived from diverse staining and imaging. The data enhancement includes transformations on hue, saturation, brightness, contrast, flipping, and shifting, as well as adding noise such as blurring and sharpening. We determined the enhancement parameters to ensure that the recognition of cervical cells will not be affected.

**Diverse data learning**. Diverse data learning on different groups of data was employed to improve the robustness of feature representations. We first trained the LR model and HR model on groups A and B with a large number of annotations to obtain baseline models. The pre-training weights on ImageNet[50] were used as the initial weights. Then we incorporated groups C and D of different styles as extra training data. Based on the baseline models, we learned more robust models on mixed data of groups A–D. The key point of our diverse data learning is the group- and category-balancing strategy. Since there are multiple groups and each group has unbalanced annotations of several subtypes, we resample the sample number of all groups and categories to make them as balanced as possible.

**Hard sample mining**. Hard sample mining was adopted to improve the accuracy of our models. For training samples that fail to be successfully classified, we conducted a second round of learning on the hard samples. The proportion of hard

samples in the entire training samples affected the tendency of models. The LR model of our recognition system was designed to find all possible lesion cells quickly and coarsely, while the HR model aimed to distinguish the true lesion cells from those candidates found by the LR model. Therefore, we used a small proportion of hard samples to train the LR model to ensure a high recall rate and a large proportion to train the HR model to ensure a high precision.

**Training details**. We employed the above three training strategies to enhance the generalization ability and recognition accuracy of the LR and HR models. According to the datasets of groups A–D in Fig. 1d, we used the train set to optimize the LR and HR models, the validation set to adjust the hyper-parameters, and then tested the performance on the test set. Groups E and F were used as independent datasets for evaluating model generalization. The positive samples were cropped around the annotations of positive slides and the negative samples were randomly cropped from negative slides. The LR and HR models used Adam[51] as the optimizer with an initial learning rate of $10^{-3}$. The strategy of learning rate decay was adopted during training.

We used a simple RNN with one hidden layer of 512 units for classifying WSIs. The RNN model was trained and validated on groups A–D and evaluated on independent groups E and F. We trained three kinds of RNNs with different inputs: the HR model features of the recommended top 10, top 20, and top 30 lesion cells. For each kind, we trained two RNNs, and then integrated the total six RNNs as our WSI classifier. In the training process, we used data augmentation to improve the varieties of the input features of the RNNs, including enhancement and rearrangement of the top $k$ lesion cell images. These strategies of model integration and data augmentation improved the robustness of the RNNs, since the limited number of slides ($<10^4$). Similarly, Adam[51] with an initial learning rate of $10^{-3}$ and the learning rate decay strategy were adopted during training the RNNs.

We deployed our system with C++ and utilized multi-threading to accelerate image processing. Our system processes one WSI under ×20 magnification in about 3.0 min using one Nvidia 1080Ti GPU. Further, we used TensorRt[36] to accelerate network model forward inference, and achieved a speed of about 1.5 min per slide. The processing speed in practice would be influenced by pixel number of slides, disk reading speed, and WSI format.

**Reporting summary**. Further information on research design is available in the Nature Research Reporting Summary linked to this article.

## Data availability

The LR model and HR model of our progressive lesion cell recognition method were initialized with the pre-training weights of ImageNet (https://www.image-net.org). We further provided the source codes, the C++ software and some test WSIs to facilitate the reproducibility. The original WSI and annotation data are private and are not publicly available since the protection of patients' privacy in cooperative hospitals. All data supporting the findings of this study are available on requests for non-commercial and academic purposes from the primary corresponding author (xlliu@mail.hust.edu.cn) within 10 working days. We do not require to sign a data use agreement. Source data are provided with this paper.

## Code availability

The source codes of this paper are available at https://github.com/ShenghuaCheng/Aided-Diagnosis-System-for-Cervical-Cancer-Screening. We also provide a C++ software and a user manual of our system with some test slides at Baidu Cloud (https://pan.baidu.com/s/1UmQzASwvlpKLO7hbwaDc_A, extracting code is *cyto*) or at Google Drive (https://drive.google.com/drive/folders/19rE9atLryIaBR8shqAlc4Sf8tn62o7ky? usp = sharing, no extracting code).

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

## Acknowledgements

We thank the members of Britton Chance Center for Biomedical Photonics for help in experiments and comments on the paper. We thank Jiangsheng Yu of Binsheng Technology (Wuhan) Co., Ltd for suggestions on the experimental design and data processing. We thank Zhongkang Medical Examination Center and Konfoong Biotech Information Ltd for data support. This work is supported by the NSFC projects (grant 61721092) and the director fund of the WNLO.

## Author contributions

X.L., J.H., L.C. and S.Z. conceived of the project. S.C., X.L., S.L. and J.Y. designed the aided diagnosis system. S.C., S.L., and J.Y. developed the algorithms. X.L., J.H., L.C., G.R., Y.X. and W.Z annotated the lesion cells, evaluated the experimental results, and completed the man–machine comparison experiments. J.H., L.C., Z.W., J.C., X.F. and F.Y. provided the cervix cytopathology glass slides. X.L., G.R., Y.X., X.L. and N.L. scanned the slides into WSIs. W.H. performed the deployment and performance optimization of the system. S.C., S.L., J.Y., X.G., J.M., X.L., Z.W. and X.Z. performed the image analysis and processing. S.C., X.L., S.L. and J.Y. wrote the manuscript. T.Q. participated in the revision of the paper. All the authors revised the paper.

## Competing interests

The authors declare no competing interests.
