## [Peer Review File · Nature Communications]

Reviewers' Comments:

Reviewer #1:

Remarks to the Author:

Overall, I really enjoyed reading this paper. Notable strength of this paper are that (1) WSI classification, (2) high specificity and sensitivity at WSI level approach with closely equivalent to the cytopathologists, and (3) high specificity and sensitivity at tile level approach.

Methodology and obtained results are sound and appropriate to conclude their study. However, at the same time, I think the technical novelty of the present method is marginal.

I believe this study is very useful for actual clinical workflow for cytopathologists and cytoscreeners in practice.

Reviewer #2:

Remarks to the Author:

The use of deep learning to improve cervical screening has a long history, and this manuscript advances progress toward automated screening of cytology slides by broadening the styles of slides that can be analyzed (via whole slide imaging) using improved training/validation/testing.

Even first generation methods (like PapNet) were sufficiently accurate in identifying completely normal slides to serve as pre-screening aids. The method gained US FDA clearance, for example, but was abandoned upon commercial consolidation with a parametric method based on defined features (NeoPath leading to FocalPoint).

Deep learning methods have advanced markedly in the past few years, so it is not surprising and is welcome to see advance. The method is still limited to distinguishing totally normal slides from any abnormality including most equivocal abnormalities (which often make up the majority of non-normal slides).

This is therefore a method that addresses the lack of cytology screeners, but not the lack of cytopathologists (or whatever skilled group classifies the severity of the abnormality, which differs by country).

In sum, this appears to be solid work, and should be interpreted as an aid to cytotechnology. I do not think the limitation is sufficiently highlighted, in its placement well into the manuscript. Whether cytology with automation will serve as a primary screening method for much of the world is increasingly doubtful. HPV testing is more sensitive as a first screen, and can be performed on self-sampled specimens without need for a speculum examination. The place of this AI-advance in a system dependent on HPV testing first is not clear, because the performance characteristics of the automated prescreen among HPV-positive women is not the subject of study. It might still be useful in dividing cytology as TRIAGE into normal vs. abnormal, to guide management. Some retraining of the algorithm might be needed, it remains to be seen.

Reviewer #3:

Remarks to the Author:

This paper presents a deep learning method to screen cervical cancer.

My main concerns are listed below.

1: The method itself is simple, all having been known.

2: The empirical evidence is subjective. First, it is based on three cytopathologists who are with limited justifications of the screening ability, which can lead to a serious bias. Second, the screening difficulty of the data used is missing, which makes me rather difficult to assess the effectiveness of this work.

3: I have not seen any new knowledge; all conclusion, results, and technique are known.

4: The only contribution is the dataset, but it will not be publicly available.

July 7, 2021

We appreciate the editor and the reviewers for their constructive comments, which are very useful for us to improve the manuscript. We carefully considered every comment and did our best to respond to them and revised the manuscript accordingly.

The point-by-point responses to the reviewers' comments are listed as below.

Reviewer #1 (Remarks to the Author):

Overall, I really enjoyed reading this paper. Notable strengths of this paper are that (1) WSI classification, (2) high specificity and sensitivity at WSI level approach with closely equivalent to the cytopathologists, and (3) high specificity and sensitivity at tile level approach.

Methodology and obtained results are sound and appropriate to conclude their study. However, at the same time, I think the technical novelty of the present method is marginal.

I believe this study is very useful for actual clinical workflow for cytopathologists and cytoscreeners in practice.

Re: We are very grateful for the evaluation of the work. At the same time, with this comment, we recognized that we did not demonstrate very clearly the novelty and major advances of our work in our original manuscript. In the current version, we tried to revise the manuscript's Introduction and Discussion to clarify more clearly the challenges facing current computer-assisted cervical cancer screening and the novelty and major advances of our work.

Revisions are listed as below:

The revised content of the Introduction is in Page 2, which is read as:

“Subject to the principles, traditional methods have low accuracy for distinguishing lesion cells with fuzzy classification boundaries and limited generalization for diverse cytology slides derived from staining and imaging.”

“However, current CNN-based computer-assisted diagnosis algorithms are insufficient in WSI-level analysis, generalization for diverse staining and imaging, and clinical-level verification. Most of the methods mainly focus on the recognition of local lesion cells, lacking WSI-level diagnostic analysis. Even a few methods³⁰⁻³¹, analyzed the whole slide cervical images on large-scale datasets, but they still did not solve the generalization problem in practical applications and clinical-level verification.”

The revised content of the Introduction is in Page 3, which is read as:

“We integrate designed data augmentation in HSV color space, diverse data learning with group and category balancing and hard sample mining to achieve high accuracy and good generalization of our system.”

“In short, our work establishes a novel WSI-level analysis system for cytopathology screening according to cervical slide characteristics of sparse-distribution and tiny-scale lesion cells and gives an effective demonstration of using deep learning to solve the bottleneck problems of current cervical screening methods. The extensive validation experiments demonstrate that our system can be used for effectively grading slides and recommending top-ranked lesion cells and reducing the workload of cytology screening staff. We believe our robust WSI analysis

system would act as an effective cytology screening assist and help accelerate the popularization of cervical cancer screening.”

The revised content of the Discussion is in Page 11, which is read as:

“Compared with the existing methods, our system has the following key technical advantages: a) a WSI-level analysis framework rather than a patch-level evaluation; b) the integrated strategy of data augmentation, diverse data learning and hard sample mining; c) the human-computer comparison verification at both tile level and WSI level; d) the practical deployment with C++, processing a giga-pixel WSI in as short as about 1.5 minutes with one GPU.”

“In recent years, WSI analysis has been widely studied in various histopathology subspecialty⁴¹⁻⁴⁵. These algorithms generally follow the below principle: first extract classification features or confidences of local tiles, then aggregate the local information to construct WSI level feature descriptors, and finally classify the slides. The lesions of histopathology slides are region-level and have overall background information. Cytopathology slides show sparse-distributed and tiny-scale lesion cells, and the cells are independent and short of overall information. These characteristics of cytopathology have brought challenges to accurate WSI analysis when directly transferring the histopathology methods. In this work, we propose a novel WSI-level analysis system for cytopathology screening according cervical slide characteristics, and demonstrate its effectiveness in classifying cytology slides by the extensive validation experiments.”

Reviewer #2 (Remarks to the Author):

The use of deep learning to improve cervical screening has a long history, and this manuscript advances progress toward automated screening of cytology slides by broadening the styles of slides that can be analyzed (via whole slide imaging) using improved training/validation/testing.

Even first generation methods (like PapNet) were sufficiently accurate in identifying completely normal slides to serve as pre-screening aids. The method gained US FDA clearance, for example, but was abandoned upon commercial consolidation with a parametric method based on defined features (NeoPath leading to FocalPoint).

Deep learning methods have advanced markedly in the past few years, so it is not surprising and is welcome to see advance. The method is still limited to distinguishing totally normal slides from any abnormality including most equivocal abnormalities (which often make up the majority of non-normal slides).

This is therefore a method that addresses the lack of cytology screeners, but not the lack of cytopathologists (or whatever skilled group classifies the severity of the abnormality, which differs by country).

In sum, this appears to be solid work, and should be interpreted as an aid to cytotechnology. I do not think the limitation is sufficiently highlighted, in its placement well into the manuscript.

Re: Yes, the reviewer is right. Our work is mainly designed to address the lack of cytology screeners, though practically it also exhibits effectiveness to some extent for cytopathologists. We highlight this limitation, and stress this method can be act as an aid to cytotechnology, and is mainly for cytology screeners.

The revised content of the Introduction is in Page 3, which is read as:

“The extensive validation experiments demonstrate that our system can be used for effectively grading slides and recommending top-ranked lesion cells and reducing the workload of cytology screening staff. We believe our robust WSI analysis system would act as an effective cytology screening assist and help accelerate the popularization of cervical cancer screening.”

We revised the expression in Discussion is in Page 10, which is read as:

“The reported positive rate of cervical cytology screening is less than 10%³⁹⁻⁴⁰, and the positive rates of the regular physical examination population are even lower. Therefore, if most of negative samples can be distinguished, it should be a great aid to cytotechnology. In our study, the distribution of slide positive scores (0.8% false negative slides) indicates that our system has potential to exclude a considerable number of negative slides and reduce a lot of cyto-screeners’ workloads. The 10/20 most suspicious areas on each slide with relatively high 88.5%/85.0% TPRs are given, which would help free doctors from the task of traversing and searching for suspicious targets and concentrate on the task of identifying recommended suspicious cells. For underdeveloped areas lacking cytopathology stuff, our system has important clinical and social significance for accelerating the popularization of cervical cancer screening.”

Whether cytology with automation will serve as a primary screening method for much of the world is increasingly doubtful. HPV testing is more sensitive as a first screen, and can be performed on self-sampled specimens without need for a speculum examination. The place of this AI-advance in a system dependent on HPV testing first is not clear, because the performance characteristics of the automated prescreen among HPV-positive women is not the subject of study. It might still be useful in dividing cytology as TRIAGE into normal vs. abnormal, to guide management. Some retraining of the algorithm might be needed, it remains to be seen.

Re: Yes, that is right whether cytology with automation will serve as a primary screening method for much of the world is increasingly doubtful.

Cervical cancer is the most common HPV-related disease (Boshartb et al. The EMBO Journal, 1984; Villier et al. The Lancet, 1987). Due to the high sensitivity and low specificity of the HPV test, and the HPV infection is usually a transient infection without any symptoms, and a positive diagnosis has a certain burden on the family and society (Schiffman, M. Papillomavirus Res, 2019), so far, cervical cytology still the mainstream cervical cancer screening method worldwide (Zhao et al. Cervical Cancer Screen and Clinical Management, Beijing Science and Technology Press, 2017).

HPV testing is indeed getting more and more attention. In 2018 US Preventive Services Task Force guideline included HPV testing alone, cotesting, and Pap testing as equal options (Curry et al. JAMA, 2018). In 2020 the American Cancer Society (ACS) published an updated guideline for cervical cancer screening (Fontham et al. Ca Cancer J Clin, 2020), to recommend cervical cancer screening with an HPV test alone every 5 years for everyone from age 25 until age 65.

So, it is interesting to see the performance characteristics of the automated prescreen among HPV-positive women. Though it is not the subject of this work, we conduct a trial on this suggestion. We analyzed the cervical cell slides from 395 HPV-positive patients (cytology positive 169, negative 226), and achieved 81.9% Specificity and 79.3% Sensitivity with 0.890 AUC (the area under ROC), 32.7% Specificity while retaining 100% Sensitivity. The results indicate that our system is still effective in samples from HPV-positive women with lower Specificity and Sensitivity than that of general cytology pap-test women. The main reason for the decreased performance is that many HPV-positive cytology-negative samples are accompanied

by bacterial infections, which increases the difficulty of classification. The trial was carried out with the ready-made networks without retraining.

This preliminary content was added in Results on Pages 6-7, which is read as,

“To demonstrate the effectiveness of our system in the case of HPV testing first or in combination with cytology, we further analyzed the cervical cell slides from 395 HPV-positive patients (cytology positive 169, negative 226). We achieved 81.9% Specificity and 79.3% Sensitivity with 0.890 AUC (Supplementary Figure 3). The preliminary results indicate that our system is still effective in the case of HPV-positive women, but the performance is obviously lower than that of general cytology pap-test women. The main reason for the decrease is that many of these HPV-positive and cytology-negative samples are accompanied by bacterial infections, which may increase the classification difficulty.”

A detailed description and a figure were added in Supplementary Page 3, which is read as,

“**The effectiveness of our system among HPV-positive women.** To demonstrate the effectiveness of our system in the case of HPV testing first or in combination with cytology, we further analyzed the cervical cell slides from 395 HPV-positive women (cytology positive 169, negative 226). These glass slides were gathered from Maternal and Child Hospital of Hubei Province and were scanned into WSIs by the second version instrument from Wuhan National Laboratory for Optoelectronics-Huazhong University of Science and Technology with 0.247 $\mu\text{m}/\text{pixel}$ under 20 \times magnification. The quantitative results of the automated prescreening among these HPV-positive women are shown in Supplementary Figure 3. Supplementary Figure 3a shows 81.9% Specificity and 79.3% Sensitivity with 0.890 AUC (the area under ROC) in case of slides from the 395 HPV-positive patient-wise WSIs. Supplementary Figure 3b shows our system achieves 32.7% Specificity while retaining 100% Sensitivity. The results indicate that our system is still effective in the case of HPV-positive women. Lower Specificity and Sensitivity than that of general cytology pap-test women (93.5% Specificity and 95.1% Sensitivity) indicates more false positive slides needed to be further reviewed by cytopathologists in HPV-positive women. The main reason for the deterioration is that many of these HPV-positive and cytology-negative samples are accompanied by bacterial infections, which may increase the classification difficulty.

Supplementary Figure 3. The quantitative results of our system among these HPV-positive women. a, The ROC of the RNN model for classifying the 395 HPV-positive patient-wise WSIs (cytology positive 169, negative 226). b, The frequency histogram of slide scores from

0 to 1 with an interval of 0.1 of the 395 slides. The blue and pink areas refer to negative and positive slides respectively.”

Reviewer #3 (Remarks to the Author):

This paper presents a deep learning method to screen cervical cancer.

My main concerns are listed below.

1: The method itself is simple, all having been known.

Re: Cervical cancer is one of the most common female cancers and caused hundreds of thousands of deaths every year worldwide, especially in developing countries. The computer-aided cervical cancer screening is important for improving the efficiency of cytopathology stuff and helping popularize cervical cancer screening. However, current computer-assisted diagnosis algorithms are insufficient in WSI-level analysis, generalization for diverse staining and imaging, and clinical-level verification. This paper focuses on the problem of robust cervical WSI analysis. To our knowledge, this question is challenging, and with the new development of deep learning (DL), we give an effective demonstration of using DL to solve this problem.

Our work establishes a novel WSI-level analysis system for cytopathology screening according cervical slide characteristics of sparse-distribution and tiny-scale lesion cells. On multi-center independent test sets of 1170 patient-wise WSIs, we achieve 93.5% Specificity and 95.1% Sensitivity for classifying slides, closely equivalent to the average level of three independent cytopathologists. The recommended top 10 lesion cells on 447 positive slides have an average TPR of 88.5%, higher and more robust than commercial Hologic ThinPrep Imaging System approved by FDA. The results demonstrate that our system would be useful for actual clinical workflow for cervical screening in practice.

The challenges facing current cervical screening algorithms were added in Introduction on Page 2, which is read as:

“Subject to the principles, traditional methods have low accuracy for distinguishing lesion cells with fuzzy classification boundaries and limited generalization for diverse cytology slides derived from staining and imaging”

“However, current CNN-based computer-assisted diagnosis algorithms are insufficient in WSI-level analysis, generalization for diverse staining and imaging, and clinical-level verification. Most of the methods mainly focus on the recognition of local lesion cells, lacking WSI-level diagnostic analysis. Even a few methods³⁰⁻³¹, analyzed the whole slide cervical images on large-scale datasets, but they still did not solve the generalization problem in practical applications and clinical-level verification.”

The challenges of cytopathology WSI analysis was added to the Discussion in Page 10, which is read as:

“In recent years, WSI analysis has been widely studied in various histopathology subspecialty⁴¹⁻⁴⁵. These algorithms generally follow the below principle: first extract classification features or confidences of local tiles, then aggregate the local information to construct WSI level feature descriptors, and finally classify the slides. The lesions of histopathology slides are region-level and have overall background information. Cytopathology slides show sparse-distributed and tiny-scale lesion cells, and the cells are independent and short of overall information. These characteristics of cytopathology have brought challenges to accurate WSI analysis when directly transferring the histopathology

methods. In this work, we propose a novel WSI-level analysis system for cytopathology screening according cervical slide characteristics, and demonstrate its effectiveness in classifying cytology slides by the extensive validation experiments.”

2: The empirical evidence is subjective. First, it is based on three cytopathologists who are with limited justifications of the screening ability, which can lead to a serious bias. Second, the screening difficulty of the data used is missing, which makes me rather difficult to assess the effectiveness of this work.

Re: That is true, the empirical evidence is subjective. It’s a standard operating procedure in cytopathology when the diagnoses of the two pathologists is inconsistent, ask a senior doctor to make an interpretation (Cibas et al. *Am J Clin Pathol*, 2001; Nakhleh et al. *Quality Improvement Manual in Anatomic Pathology*. College of American Pathologists, 2002). In this work, we produced the final annotations by a consensus of three cytopathologists. For most typical cervical cells, cytopathologists’ interpretation is usually consistent. But for some atypical lesion cells such as ASC-US, the inconsistency of interpretation by different doctors probably occurs. Generally, the laboratory uses a quality control system to ensure that the overall proportion is within a certain range, and fluctuations of this type of slide are clinically permitted (Nayar et al. *The 2014 Bethesda System for Reporting Cervical Cytology*. Springer, 2015).

That label noise challenges the cytology WSI analysis, so the data screening is a routine pretreatment step before deep learning training, and we added the data screening procedure in the revised manuscript in the Methods on Page 13, which is read as:

“Based on the TBS criteria⁶, the cervical slides are annotated by 6 cytopathologists using Qupath⁴⁷ and a home-made Semi-automatic online annotation software. Considering that it’s a standard operating procedure in cytopathology when the diagnoses of the two pathologists is inconsistent, ask a senior doctor to make an interpretation⁴⁸⁻⁴⁹. In this work, the final annotations produced were by a consensus of three cytopathologists. We abandoned questionable annotations. Based on the TBS criteria⁶, we classify all kinds of Squamous Epithelial Cell Abnormalities (including Atypical Squamous Cells (ASC), Squamous Intraepithelial Lesion (SIL), and Squamous cell carcinoma (SCC)) and Glandular Epithelial Cell Abnormalities (including Atypical Endocervical/Glandular cell, and Endocervical adenocarcinoma in situ (AIS), and Adenocarcinoma) as positive labels, and classify various normal cellular elements, nonneoplastic finding such as Nonneoplastic cellular variations, reactive cellular changes, and glandular cells status post hysterectomy as negative labels NILM (Negative for Intraepithelial Lesion or Malignancy). During the iterative learning of our system, we collected massive false negatives of various morphologies as negative annotations.”

3: I have not seen any new knowledge; all conclusion, results, and technique are known.

Re: The core goal of this work is to use the increasingly developed deep learning methods to solve the problems of cervical cell screening. To our knowledge, this question is still challenging regarding WSI-level analysis, generalization for diverse staining and imaging, and clinical-level verification. In this paper, we analyze cervical slide characteristics and develop a robust cervical WSI analysis method and give an effective demonstration of using DL to solve this challenging problem.

We validate our system on multi-center independent test sets of 1,170 patient-wise WSIs in two kinds of clinical-level diagnosis analysis: WSI classification and top lesion cells recommendation. On the level of WSI classification, we achieve 93.5% Specificity and 95.1%

Sensitivity for classifying slides, closely equivalent to the average level of three independent cytopathologists. On the level of top lesion cells recommendation, compared with current commercial Hologic ThinPrep Imaging System, our system has a higher TPR of recommended cells and can be applied to WSIs of diverse staining and imaging styles. To the best of our knowledge, this is the first clinical-level multi-center results of assisted cervical cancer screening. The results demonstrate that our work successfully establishes robust whole slide image analysis system for cervical cancer screening and lay a foundation for the practicability in diverse data scenarios.

In the revised manuscript, we expressed more clearly our efforts to address the main challenges of current computer assisted cervical cancer screening, our method's advances, and contributions to the field of computer assisted cervical cancer screening. The revisions in Introduction and Discussion are listed as below.

The conclusion of our work in Introduction is on Page 3, which is read as:

“In short, our work establishes a novel WSI-level analysis system for cytopathology screening according cervical slide characteristics of sparse-distribution and tiny-scale lesion cells and give an effective demonstration of using deep learning to solve the bottleneck problems of current cervical screening methods. The extensive validation experiments demonstrate that our system can be used for effectively grading slides and recommending top-ranked lesion cells and reducing the workload of cytology screening staff. We believe our robust WSI analysis system would act as an effective cytology screening assist and help accelerate the popularization of cervical cancer screening.”

The challenges of current cervical screening algorithms were added in Introduction on Page 2, which is read as:

“Subject to the principles, traditional methods have low accuracy for distinguishing lesion cells with fuzzy classification boundaries and limited generalization for diverse cytology slides derived from staining and imaging”

“However, current CNN-based computer-assisted diagnosis algorithms are insufficient in WSI-level analysis, generalization for diverse staining and imaging, and clinical-level verification. Most of the methods mainly focus on the recognition of local lesion cells, lacking WSI-level diagnostic analysis. Even a few methods³⁰⁻³¹, analyzed the whole slide cervical images on large-scale datasets, but they still did not solve the generalization problem in practical applications and clinical-level verification.”

The characteristics of cervical WSIs were added in Discussion Page 11, which is read as:

“In recent years, WSI analysis has been widely studied in various histopathology subspecialty⁴¹⁻⁴⁵. These algorithms generally follow the below principle: first extract classification features or confidences of local tiles, then aggregate the local information to construct WSI level feature descriptors, and finally classify the slides. The lesions of histopathology slides are region-level and have overall background information. Cytopathology slides show sparse-distributed and tiny-scale lesion cells, and the cells are independent and short of overall information. These characteristics of cytopathology have brought challenges to accurate WSI analysis when directly transferring the histopathology methods. In this work, we propose a novel WSI-level analysis system for cytopathology screening according cervical slide characteristics, and demonstrate its effectiveness in classifying cytology slides by the extensive validation experiments.”

The technical advantages of our work were added in Discussion Page 11, which is read as:

“Compared with the existing methods, our system has the following key technical advantages: a) a WSI-level analysis framework rather than a patch-level evaluation; b) the integrated strategy of data augmentation, diverse data learning and hard sample mining; c) the human-computer comparison verification at both tile level and WSI level; d) the practical deployment with C++, processing a giga-pixel WSI in as short as about 1.5 minutes with one GPU.”

4: The only contribution is the dataset, but it will not be publicly available.

Re: In this work, we establish a robust whole slide image analysis framework to solve the bottleneck problems of current assisted cervical screening. The big data (WSIs and annotation lesion cells), problem analysis, the method establishment, and the validation are all vital to problem solving. At present, the WSIs and annotation data are private and are not publicly available since the protection of patients' privacy in cooperative hospitals. We provide the source data of all figures and tables in the manuscript to prove the credibility and reproducibility of our method. After obtaining the permission of the cooperative hospital and removing patients' sensitive information, we provided a small number of slides for testing our software.

Reviewers' Comments:

Reviewer #2:

Remarks to the Author:

To achieve 100% sensitivity (if that is the goal) two thirds of HPV positive women would be referred.

I think the work is interesting, but the performance for triage is not established as sufficient for management of HPV-positive women.

Reviewer #3:

Remarks to the Author:

This is a revised submission. I think that is ready for publication. My concerns have been addressed properly, so I would like to recommend to accept.